# Escaping Collapse: The Strength of Weak Data for Large Language Model Training[*]

**Kareem Amin**  **Sara Babakniya**[†]  **Alex Bie**

**Weiwei Kong**  **Umar Syed**  **Sergei Vassilvitskii**

Google Research

## Abstract

Synthetically-generated data plays an increasingly larger role in training large language models. However, while synthetic data has been found to be useful, studies have also shown that without proper curation it can cause LLM performance to plateau, or even "collapse", after many training iterations. In this paper, we formalize this question and develop a theoretical framework to investigate how much curation is needed in order to ensure that LLM performance continually improves. Our analysis is inspired by boosting, a classic machine learning technique that leverages a very weak learning algorithm to produce an arbitrarily good classifier. The approach we analyze subsumes many recently proposed methods for training LLMs on synthetic data, and thus our analysis sheds light on why they are successful, and also suggests opportunities for future improvement. We present experiments that validate our theory, and show that dynamically focusing labeling resources on the most challenging examples — in much the same way that boosting focuses the efforts of the weak learner — leads to improved performance.

## 1 Introduction

Large Language Models (LLMs) represent the frontier of artificial intelligence, and are trained on vast amounts of human-generated data. However, much of the high-quality publicly available data on the Internet has been exhausted, and limits on generating new tokens threaten to slow progress on LLM training.

As a consequence, synthetically-generated datasets are playing an important role in the training of LLMs. Synthetic data have been shown to improve the performance of real large models on a range of tasks [BKK+22, ZWMG22, GPS+23, SCA+24]. On the other hand, the circuitous nature of training new LLMs on data generated by previous generations of LLMs has caused concerns of model collapse [SSZ+24, ACRL+24].

What makes synthetic data beneficial or harmful? The answer depends on the precise elements of the synthetic data recipe, and one of our main contributions is a theoretical framework that unifies existing elements of synthetic data approaches, facilitating reasoning about when they might succeed or fail.

---

[*]Authors ordered alphabetically. Author contributions are listed at the end of the paper. Correspondence to {kamin,babakniya,alexbie,weiweikong,usyed,sergeiv}@google.com.

[†]Work done while the author was a student at the University of Southern California.

39th Conference on Neural Information Processing Systems (NeurIPS 2025).

Basic learning theory and empirical studies suggest that a necessary condition for avoiding model collapse is that synthetic data is curated in some way to inject signal that is exogenous to the system that produced the original data. This can come in many forms: identification of high-quality subsets of synthetic data, human rewrites of poor responses, a separate model rating the responses, *etc*. A key question is how much curation is *sufficient* to not only avoid collapse, but also to converge to an optimal LLM? Our answer, which we will make precise, is the minimum amount.

Specifically, we analyze a simple procedure for improving an LLM, in which we iteratively (1) generate synthetic responses from the model; (2) obtain additional responses from an exogenous source; and (3) train the next generation of the model with both types of responses. This procedure captures previous successful approaches for training LLMs on synthetic data [ZWMG22, GPS$^+$23, SCA$^+$24], and so our analysis provides an explanation for why they work. More broadly, it models the ad hoc processes employed by model developers. We show that if at least a $\beta > 0$ fraction of the non-synthetic responses (*i.e.*, the ones produced by an external signal) are correct, then the iterative procedure converges to an optimal LLM (*i.e.*, one that returns a correct response to each prompt). See Theorem 5 for the precise statement and exact convergence rate.

**Connection to Boosting.** At a high level, our analysis shows how to use synthetic data to focus curation on regions of the prompt space where the models perform poorly. In this way, this approach resembles AdaBoost, a classic machine learning algorithm that iteratively focuses a weak learning algorithm on training examples where previous weak hypotheses performed poorly. Unlike boosting, however, our assumptions on the data and the learning method are inverted. Instead of a weak learner, we assume access to powerful LLMs that can perfectly model an input distribution, which we call *strong learners.* However, we also assume access to only weak information about the distribution we wish to model (specifically, that $\beta > 0$), i.e. *weak data.* This is in contrast to traditional boosting where the algorithm has access to *strong data*, i.e., independent and identically distributed (i.i.d.) examples from some target distribution.

This connection between the theory of boosting and learning from synthetic data has been largely unexamined in the existing literature. Our analysis also suggests practical ways to improve current algorithms for learning from synthetic data. In our experiments, we show that scarce curation resources are better utilized by focusing their efforts on producing responses to the most challenging prompts in the training set.

## 2    Related work

Training models on human-generated data only has limitations such as scalability, biases, errors, and potential privacy considerations [KPS$^+$23, SCA$^+$24, GAK23, LWX$^+$24]. [LML$^+$24] highlights a challenge: as LLMs scale, the demand for high-quality data increases, yet access to such data becomes more restricted due to copyright and privacy constraints. Given these challenges, integrating synthetic data into training pipelines is essential but comes with risks.

**Model Collapse.** Several studies highlight a critical concern regarding the use of synthetic data in training LLMs, known as model collapse. This phenomenon is caused by *improper* use of synthetic data in training the model, which can cause performance degradation or even complete failure of the model [SKAK25]. [ACRL$^+$24, SSZ$^+$24, HBA23, GSD$^+$24] have empirically studied model collapse in various settings, demonstrating the detrimental effects of iterative training on only synthetic data and highlighting how this process can severely degrade model performance.

[DFK24, DFY$^+$24, BBD$^+$24, DD24, SCH$^+$24] study model collapse theoretically. Their results show that recursively retraining only on synthetic data causes performance degradation in different models. However, combining synthetic and labeled training data [BBD$^+$24, DD24, SCH$^+$24, KSD$^+$24, FBBG24] can mitigate this performance degradation. In contrast to our work, they do not demonstrate continuous improvement toward an optimal model.

Recently, [STK24] and [FDY$^+$24] provide theoretical explanations for model collapse under restricted models, including Gaussian mixture models and linear classifiers. [FDY$^+$24] shows that the presence of a verifier to select more desired synthetic data can improve the performance in non-recursive settings. [FSH$^+$24] extends this work to include both noisy labels and features and provide theoretical and empirical results on the impacts of combining synthetic and real data. Our results do not assume a specific learning class, instead relying on a black-box strong learning assumption.

**Self Improving LLMs** Self-evolving or self-improving LLMs [TLC+24] is a new research direction that leverages the model itself to generate or guide the creation of high-quality data [WKM+23, HGH+23, GFA+24], which can then be used for fine-tuning [YPC+24, CDY+24] or RLHF [PMM+24], enabling continuous improvement with minimum or no external intervention.

STaR [ZWMG22] presents a bootstrapping mechanism to enhance the reasoning capabilities of LLMs by iteratively asking the model to generate step-by-step "chain-of-thought" rationales for questions, filtering out incorrect answers, fine-tuning the original model on all correct rationales, and repeating the process. ReST [GPS+23] proposes a combination of self-generated data and offline reinforcement learning. The method operates iteratively in two primary phases: a "Grow" phase, where for each input (context), the LLM generates multiple outputs to expand the training dataset, and an "Improve" phase, which involves ranking and filtering this augmented dataset using a learned reward model trained on human preferences. ReST$^{EM}$ [SCA+24] is a modified version of ReST with two main differences; they do not augment the generated data with human-generated data, and in the "Improve" step instead of fine-tuning the model in the previous iteration, they fine-tune the base model. All of the above methods can be modeled in our framework, and thus we provide a better theoretical understanding about why and when such methods can work.

Recent works [SZE+25, YFC+25, DDE+24] take a more theoretical approach to understand self-improving algorithms. [SZE+25] introduces a new metric to analyze how different components contribute to self-improvement formally. In parallel, [YFC+25] studies how to optimally allocate a fixed computational budget across iterations of synthetic data generation and fine-tuning, showing that exponential growth policies outperform constant or linear ones in both theory and practice.

## 3 Preliminary Notation

**Datasets.** Let $\mathcal{X}$ be the set of all possible *prompts*, and let $\mathcal{Y}$ be the set of all possible *responses*, which we also call *labels*. An element of $\mathcal{X} \times \mathcal{Y}$ is a *labeled prompt*. A subset of $\mathcal{X}$ is a *prompt set*, and a subset of $\mathcal{X} \times \mathcal{Y}$ is a *dataset*.

For any prompt set $P$, let $P(x)$ denote the number of times prompt $x$ appears in $P$, and for any dataset $D$, let $D(x, y)$ denote the number of times labeled prompt $(x, y)$ appears in $D$. Typically we have $P(x) \in \{0, 1\}$ and $D(x, y) \in \{0, 1\}$. However, we also allow datasets to contain multiple copies of the same element, where the multiplicity, or *weight*, of an element can be any non-negative real number, i.e., $D(x, y) \in \mathbb{R}_+$. We write $(x, y) \in D$ if and only if $D(x, y) > 0$ and $|D| = \sum_{x,y} D(x, y)$. Datasets with general weights are formed by using the weighted union operation: If $D_0$ and $D_1$ are datasets, and $\lambda_0, \lambda_1 > 0$, then $D = \lambda_0 D_0 \uplus \lambda_1 D_1$ is the dataset defined by $D(x, y) = \lambda_0 D_0(x, y) + \lambda_1 D_1(x, y)$.

For any dataset $D$ let $D(y|x) = D(x, y)/\sum_{y'} D(x, y')$ be the fraction of labeled prompts in $D$ with prompt $x$ that have response $y$. Define $D(y|x) = 0$ if $\sum_{y'} D(x, y') = 0$.

A large language model, or *LLM*, is a function that maps each prompt in $\mathcal{X}$ to a distribution on $\mathcal{Y}$. We will denote LLMs by the symbol $g$, and let $g(x)$ denote the distribution over labels $\mathcal{Y}$ of $g$ when evaluated on prompt $x$.

Let $q : \mathcal{X} \times \mathcal{Y} \to \{0, 1\}$ be the *quality* function, where $q(x, y) = 1$ indicates that $y$ is a good response to prompt $x$.

## 4 Problem Setting

We consider a setting where a sequence of LLMs $g_1, g_2, \ldots$ are learned on a sequence of datasets $\mathcal{D}_1, \mathcal{D}_2, \ldots$. Given a prompt set $P$, our high-level goal is to produce an LLM that generates high quality responses for every prompt in $P$. We illustrate this meta-algorithm in Setting 1.

Unlike classical learning, where the learner has access to samples from the target distribution, we assume that the learner only has access to labeled examples constructed by a data generation procedure that we control, denoted by the function `GenerateData`. Data generation might make use of synthetic data, produced by the previous generation's LLM $g_{t-1}$, and exogenous (*i.e.*, non-synthetic) signals.

In order to formalize our goal, we make precise the capabilities of `learner`, the capabilities of `GenerateData`, and our notion of quality.

---

**Setting 1** Data Generation Problem

---

    **Given:** Prompt set $P$, number of iterations $T$.

1: $g_0 = \bot$
2: **for** $t = 1, 2, \ldots, T$ **do**
3:     $D = \texttt{GenerateData}(P, g_{t-1})$
4:     $\mathcal{D}_t = \mathcal{D}_{t-1} \uplus D$
5:     $g_t = \texttt{learner}(\mathcal{D}_t)$
6: **end for**
7: Output $g_T$.

---

## 4.1 Strong Learning

We first introduce the concept of a strong learner.

**Definition 1** (Strong Learner). For any LLM $g$ let $g(y|x)$ be the probability that the distribution $g(x)$ assigns to response $y$. The function `learner` takes as input a dataset $D \subset \mathcal{X} \times \mathcal{Y}$ and outputs an LLM $g$ such that $g(y|x) = D(y|x)$ for all $(x, y) \in D$.

The procedure `learner` trains an LLM that matches the conditional probability of each response given a prompt in the input dataset. That is, we assume that the model class has the capacity to match this distribution exactly, and the learning procedure can find the model parameters that perfectly fit the data. This assumption is motivated by the fact that deep neural networks instantiate all modern LLMs and are both theoretically capable of approximating arbitrary functions [MP99] and frequently observed to fit their training inputs [ZBH+21].

While LLMs are powerful, the largest models contain billions of parameters and are extremely expensive to train. Thus, training $T$ state-of-the art models from scratch is prohibitively expensive. In contrast, given model $g_{t-1}$ trained on $\mathcal{D}_{t-1}$, it is significantly less expensive to train a model $g_t$ on $D_{t-1} \uplus D$, for some choice $D$. In other words, our setting models continued training, where the training mixture for the next LLM is constructed by augmenting the existing data mixture with new examples.

## 4.2 Data Generation

Creating data for the next generation of an LLM might involve making use of synthetic data produced by the current generation of the LLM. To avoid model collapse, some degree of data curation happens in practice. This curation may make use of an exogenous signal previously unknown to our training algorithm. Curation may also take the form of evaluating the quality of existing synthetic data. We discuss each of these capabilities in greater detail.

**Synthetic Data.** Given an LLM $g$, and a prompt $x$, we can generate a synthetic response for $x$ by sampling from distribution $g(x)$. Overall, we assume that synthetic data generation is relatively inexpensive, and permit data generation procedures that make calls to previously-trained LLMs.

**Noisy Filter.** We assume that we can noisily partition a dataset into labeled and unlabeled prompts based on the quality function.

**Definition 2** ($\gamma$-noisy Filter). Let $\gamma \in [0, 1]$. The function $\texttt{filter}_\gamma(D)$ takes as input a dataset $D \subseteq \mathcal{X} \times \mathcal{Y}$ and outputs $(S^+, P^-) = \texttt{filter}_\gamma(D)$, where the following holds with independent probability for each $(x, y) \in D$:

- If $q(x, y) = 1$ then $(x, y) \in S^+$ with probability at least $\gamma$, and otherwise $x \in P^-$.

- If $q(x, y) = 0$ then $x \in P^-$.

In other words, $\gamma$ is a lower bound on the recall of $\texttt{filter}_\gamma$ for recognizing high-quality responses. For many applications, recognizing that a synthetic response is a high-quality for a given prompt is significantly easier than generating the response from scratch. For instance, if the dataset contains arithmetic or coding problems, it is relatively easy to programmatically verify a correct answer.

**Weak Labeler.** Key to our work is the notion of a *weak labeler*, a function, that given any set of prompts produces responses with average quality bounded away from zero.

To formally define it, we use an auxiliary function $a_P : P \to \mathcal{Y}$, which generates labels for all prompts in a set $P$.

**Definition 3** ($\beta$-weak Labeler)**.** Let $\beta \in [0, 1]$. The function $\texttt{labeler}_\beta$ takes as input a prompt set $P \subset \mathcal{X}$, and uses an auxiliary function $a_P : \mathcal{X} \to \mathcal{Y}$ to label every prompt in $P$. Formally,

$$\texttt{labeler}_\beta(P) = \{(x, y) : x \in P, y = a_P(x)\} \subset \mathcal{X} \times \mathcal{Y}$$

We say that the labeler is $\beta$-weak if a $\beta$ fraction of these labels are high-quality, i.e., for any input prompt set $P$,

$$\frac{|\{(x, y) \in \texttt{labeler}_\beta(P) : q(x, y) = 1\}|}{|P|} \geq \beta.$$

In our setting, each iteration of data generation is allowed to make one call to the weak labeler. The role of the labeler is to create new responses to a set of prompts. We are not prescriptive about how the labeler is implemented, only that it provides some $\beta$ fraction of high-quality responses. The labeler does not need to indicate *which* prompts have been correctly labeled, nor does it need to correctly label a representative portion of its input. For example, the labeler is allowed to only correctly label the "easiest" prompts that it receives as input. We think of these responses as being produced by an exogenous process, such as consulting with a human directly, having a human correct or critique LLM responses, or any other framework for generating responses that are not purely synthetic.

### 4.3 Objective

Given these capabilities — the ability to synthesize data, assess synthetic data quality, and weakly label new data — the **goal** of our algorithm is to construct datasets $\mathcal{D}_1, \ldots, \mathcal{D}_T$ so that

$$\lim_{T \to \infty} \Pr_{x \sim P, y \sim g_T(x)}[q(x, y) = 1] = 1 \tag{1}$$

where $x \sim P$ denotes that $x$ is chosen uniformly at random from $P$, and $y \sim g_T(x)$ denotes that $y$ is chosen from distribution $g_T(x)$. In other words, as the number of algorithm iterations grows large, the final LLM output by the algorithm returns a correct response to almost every prompt in $P$. Note that this objective is similar to the objective of classical boosting. Rather than use weak learners to construct a good hypothesis, we ask whether *strong learners* and *weak data* can be used to construct a model that provides high-quality results on all prompts.

## 5   Algorithm

We present an algorithm for learning an LLM from a mixture of synthetically generated and weakly labeled data that uses the capabilities introduced in Section 4.

The aforementioned algorithm generates synthetic responses from the last generation of LLM. Synthetic data generation is given multiple opportunities to produce good responses, which are noisily recognized by $\texttt{filter}_\gamma$. Prompts that are consistently paired with low-quality responses are passed into $\texttt{labeler}_\beta$, which provides a minimal amount of signal. A mixture of good synthetically labeled data and $\beta$-weak-labeled data is then incorporated into the training mixture. To state this procedure formally, it will be convenient to introduce the $\texttt{generate}$ subroutine, which issues multiple calls to an LLM per prompt to produce a dataset of synthetically labeled prompts.

**Definition 4** (Generation)**.** The function $\texttt{generate}$ takes as input a prompt set $P \subseteq \mathcal{X}$, LLM $g$ and positive integer $k$, and is defined

$$\texttt{generate}(P; k, g) = \{(x, y_x^i) : x \in P, i \in [k], y_x^i \sim g(x)\}.$$

Algorithm 2 formalizes our procedure for data generation, where generation, filtering, and weak-labeling are applied in sequence on each generation of LLM. Whether the data that is being added to the mixture consists of mostly $\beta$-weakly labeled data ($D_t$ in Algorithm 2) or $\gamma$-filtered synthetic data ($S_t^+$ in Algorithm 2) is parameterized by $\alpha > 0$.

---

**Algorithm 2** Boosting-style algorithm for LLM training

---

**Given:** Prompt set $P$, repeat parameter $k$, weakly labeled prompt weight $\alpha$, high-quality fraction $\beta$, filter recall $\gamma$, number of iterations $T$.

1: $g_0 = \bot$ and $\mathcal{D}_0 = \emptyset$        ▷ Initial LLM and initial training set
2: **for** $t = 1, 2, \ldots, T$ **do**
3:     $S_t = \texttt{generate}(P; k, g_{t-1})$        ▷ Generate $k$ synthetic responses per prompt.
4:     $(S_t^+, P_t^-) = \texttt{filter}_\gamma(S_t)$        ▷ Noisily partition high-quality synthetic data from low-quality prompts.
5:     $D_t = \texttt{labeler}_\beta(P_t^-)$        ▷ Weakly label low-quality prompts
6:     $\lambda_t = \frac{\alpha}{|D_t|}$        ▷ Set weight of weakly labeled prompts
7:     $\mathcal{D}_t = \mathcal{D}_{t-1} \uplus \lambda_t D_t \uplus S_t^+$        ▷ Update training mixture
8:     $g_t = \texttt{learner}(\mathcal{D}_t)$        ▷ Use learner to update LLM
9: **end for**

---

## 6 Main result

Theorem 5 is our main theoretical result, and states that the final LLM $g_T$ output by Algorithm 2 satisfies the convergence requirement in Eq. (1). Theorem 5 also quantifies the rate of convergence.

**Theorem 5.** *Let $\varepsilon \in (0, 1)$. Suppose that in Algorithm 2 we have $\alpha > 0$, $\beta \in (0, 1)$, $\gamma \in (0, 1]$*

$$T \geq \frac{\log(2/\varepsilon)}{\beta} + \frac{2\alpha}{\beta\varepsilon} + 1$$

*and $k \geq (2 \log T + \log |P|)/(\beta\gamma)$. With probability at least $1 - 1/T$ over the randomness of the algorithm, the final LLM $g_T$ output by the algorithm satisfies*

$$\Pr_{x \sim P, y \sim g_T(x)}[q(x, y) = 1] \geq 1 - \varepsilon.$$

*Note that by setting $\alpha = \varepsilon$ in Algorithm 2 the iteration complexity becomes $T = O(\log(1/\varepsilon)/\beta)$.*

*Proof sketch.* The key step in the proof is showing that, with probability $1 - 1/T$, in each iteration $t$ we have $\Pr_{y \sim g_{t-1}(x)}[q(x, y) = 1] \geq \beta$ for all but $(1 - \beta)^{t-1}$ fraction of the prompts $x \in P$. Since the algorithm draws $k = \Omega(1/(\beta\gamma))$ synthetic responses to each prompt from $g_{t-1}$, one of those responses is likely to be correct. As a result, correctly labeled prompts are continually added to the training data (via the synthetic dataset $S_t^+$), and the quality of the training data steadily improves, causing the performance of the LLMs learned from that training data to approach the optimal performance.

Even when $t$ is large, it is non-trivial to show that $\Pr_{y \sim g_{t-1}(x)}[q(x, y) = 1] \geq \beta$ for nearly all prompts $x \in P$. While this fact follows from our assumption about the weak labeler, it does not follow straightforwardly. The weak labeler ensures that the *average* response quality to a given set of prompts is at least $\beta$, but we need a guarantee about response quality that holds *uniformly* for almost all prompts. Our approach is to first show that $P_t^-$ (the set of prompts with low-quality responses) shrinks exponentially with $t$, and then observe that the total weight assigned to these prompts in the training data is fixed at $\alpha > 0$ (a free parameter of our algorithm). Consequently, once a prompt is assigned a high-quality response by the weak labeler, the weight of that response overwhelms the weight of all previous low-quality responses in the training data. So when the learner fits an LLM to this training data, the LLM assigns non-trivial probability mass to the high-quality response; we are able to bound this probability from below by $\beta$.     $\square$

### 6.1 Relationship to Boosting

Boosting is a meta-learning algorithm for combining weak hypotheses into highly accurate ensemble classifiers [SF13]. While the most common version of boosting is AdaBoost [FS97], we will present a slightly simpler version that still contains all of the essential ideas.

In each iteration of boosting, a training set of binary-labeled examples is given as input to a *weak learner*. Each training example is associated with a non-negative weight, and the weights sum to 1. The weak learner returns a hypothesis that achieves weighted error at most $\frac{1}{2} - \beta$ on the training set,

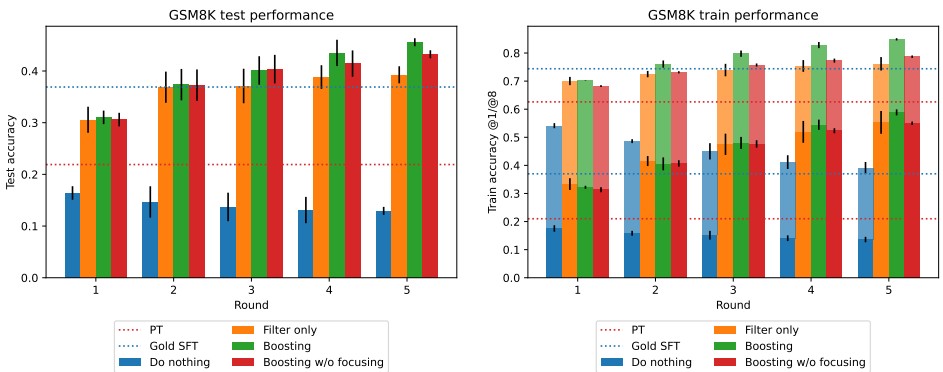

Figure 1: We plot test and train performance of our Algorithm 2 variants on GSM8K, across rounds. We report the mean and `np.std(*,ddof=1)` for 3 seeds. For train accuracy plots, we plot both train accuracy@1 (solid) and train accuracy@8 (stacked). Boosting results displayed here use *weak data (A)*.

where $\beta \in (0, \frac{1}{2})$ is the *edge* over the trivial hypothesis that randomly guesses each label. The weight on each training example that is correctly labeled by the hypothesis is decreased by a factor $\exp(\theta)$, and the weight on each training example that is incorrectly labeled by the hypothesis is increased by the same factor, where $\theta = \frac{1}{2} \log \frac{1+2\beta}{1-2\beta}$. Essentially, the weights are adjusted to concentrate on difficult examples. The weights are renormalized to sum to 1, and the process repeats. After $T = O(\log(1/\varepsilon)/\beta^2)$ iterations, a majority vote among all the hypotheses achieves unweighted error most $\varepsilon$ on the training set.

Comparing Algorithm 2 to the description of boosting given above reveals many similarities. In each iteration of Algorithm 2, prompts are given as input to a weak labeler that has quality $\beta \in (0, 1)$, where $\beta$ is the edge over the trivial labeler that assigns an incorrect response to every prompt. The weight on each prompt that is correctly labeled by the previous iteration's LLM is set to zero, and the weight on each prompt that is incorrectly labeled by the previous iteration's LLM is increased by at least a factor $\exp(\theta)$, where $\theta = \log \frac{1}{1-\beta}$ (this fact emerges from our analysis, which proves that size of the set of prompts given to the weak labeler shrinks by a factor at least $1 - \beta$ each iteration; see Lemma 11(b) in the Appendix). As in boosting, the weights are adjusted to concentrate on difficult examples. After $T = O(\log(1/\varepsilon)/\beta)$ iterations, an LLM learned from all of the training data achieves error at most $\varepsilon$ on the overall prompt set (see Theorem 5).

## 7 Experiments

Viewing Algorithm 2 as a meta-algorithm, we conduct experiments with specific instantiations using Gemma 2 2B on math problem solving [CKB[+]21, GSM8K] and Python coding [AON[+]21, MBPP] tasks.[3] We select these tasks because measures of response quality here are consistent and easily verifiable.

### 7.1 Instantiations of Algorithm 2

**Do nothing.** Responses produced by the current iteration of the model are directly used as training data for the next iteration. This corresponds to setting $\alpha = 0$, omitting the `generate` operation, and using a pass-through `filter` in line 4 of Algorithm 2. This tracks the setting explored in the "model collapse" literature [ACRL[+]24, SSZ[+]24, GSD[+]24].

**Filter only.** Only correct responses in the current iteration are used for training in the next iteration. This corresponds to $\alpha = 0$ and $\gamma = 1$ in Algorithm 2. This reproduces the STaR/ReST approaches for learning from synthetic data [ZWMG22, GPS[+]23, SCA[+]24].

---

[3]Gemma models are made available under Google's Gemma Terms of Use. GSM8K and MBPP are made available under the MIT License.

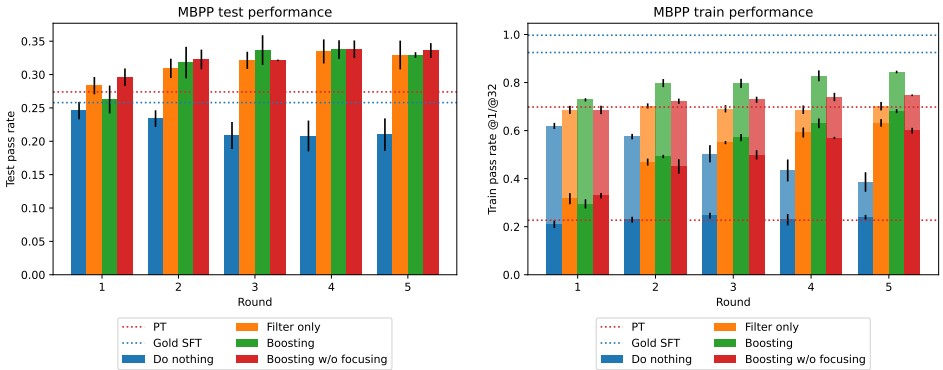

Figure 2: We plot test and train performance of our Algorithm 2 variants on MBPP, across rounds. We report the mean and `np.std(*,ddof=1)` for 3 seeds. For train pass rate plots, we plot both train pass@1 (solid) and train pass@32 (stacked). Boosting results displayed here use *weak data (A)*.

**Boosting.** The full algorithm of the present paper. In addition to the synthetic data produced by *Filter only*, we mix in weak data from the labeler. This corresponds to $\alpha > 0$ and $\beta > 0$ in Algorithm 2. We use $\alpha = 1/3$ in all experiments.

- **Boosting, w/o focusing**. We ablate out focusing on hard examples. To be precise: rather than giving the labeler the prompts we got wrong, $P_t^-$, we draw a random set of questions of size $|P_t^-|$.

We also report two baselines that do not involve iteratively training on model-generated data. **PT**: the pre-trained model; and **Gold SFT**: the model after one round of fine-tuning on the human-written responses in the dataset. Note that *Gold SFT* is the only setup that makes use of human-written responses, rather than just for answer verification.

## 7.2 Experimental Details

In all experiments, a round of fine-tuning entails training all parameters of the model for 330 (GSM8K) or 30 (MBPP) steps at batch size 64 (with the exception of *Gold SFT* where we report the checkpoint with best validation accuracy) We train with standard sequence cross-entropy loss. Training examples are (`input`, `target`) pairs, where `input` is the problem preceded by a 3-shot prompt (see Appendix D for prompt templates); and `target` is a model response (human-written response for *Gold SFT*).

**Modeling the weak data.** We instantiate `labeler` as a Gemma 2 2B PT model with a fixed *total query budget*, which is distributed uniformly over all problems it receives. For a given problem, we sample responses from the model equal to that problem's allotted queries. We consider two setups to simulate weak data provided by the weak labeler. **Weak data (A)**: for each question, we return all correct responses if there are any. If there are none, we return a random incorrect response. **Weaker data (B)**: we pool together the correct responses to all questions. We add to this collection an equal number of incorrect responses, drawn randomly from all incorrect responses to all questions.

We remark that the fixed total query budget setup offers a mechanism for satisfying the weak data assumption: the labeler can maintain constant accuracy when targeting increasingly granular (and more difficult) slices of the input distribution by focusing their resources. We see that this is indeed the case experimentally, and plot accuracies in Figure 3. Moreover, a fixed query budget is a natural analogue to the fixed person-hours/money/compute budgets behind a labelling effort.

**Departure from the theory.** In our experiments, we make one main modification from Algorithm 2. Rather than accumulating *data* and retraining the model each iteration (Algorithm 2, line 9), we instead accumulate *updates*. That is, we fine-tune on the newly introduced data in each iteration, initializing from the checkpoint produced by the prior iteration. We do this for efficiency reasons.

| Setup | Rounds | GSM8K | | | MBPP | | |
|---|---|---|---|---|---|---|---|
| | | train | | test | train | | test |
| | | @1 | @8 | greedy | @1 | @32 | greedy |
| PT 
 - orig. report | 0 | $.211_{.002}$ 
 - | $.630_{.007}$ 
 - | $.222_{.003}$ 
 $.243*$ | $.235_{.008}$ 
 - | $.703_{.007}$ 
 - | $.275_{.002}$ 
 $.302*$ |
| Gold SFT | 1 | $.392_{.020}$ | $.755_{.009}$ | $.379_{.008}$ | $.880_{.039}$ | $.987_{.009}$ | $.237_{.018}$ |
| Do nothing | 5 | $.136_{.012}$ | $.391_{.026}$ | $.129_{.022}$ | $.238_{.011}$ | $.386_{.041}$ | $.210_{.024}$ |
| Filter only | 5 | $.553_{.050}$ | $.762_{.029}$ | $.393_{.020}$ | $.632_{.016}$ | $.702_{.018}$ | $.329_{.022}$ |
| Boosting **(A)** 
 - w/o focusing | 5 | $.589_{.013}$ 
 $.550_{.008}$ | $.849_{.005}$ 
 $.787_{.005}$ | $.456_{.010}$ 
 $.432_{.013}$ | $.681_{.008}$ 
 $.600_{.012}$ | $.844_{.005}$ 
 $.747_{.003}$ | $.329_{.004}$ 
 $.336_{.011}$ |
| Boosting **(B)** 
 - w/o focusing | 5 | $.565_{.020}$ 
 $.509_{.000}$ | $.820_{.009}$ 
 $.767_{.007}$ | $.443_{.020}$ 
 $.430_{.012}$ | $.647_{.012}$ 
 $.544_{.012}$ | $.832_{.009}$ 
 $.698_{.010}$ | $.326_{.009}$ 
 $.327_{.009}$ |

Table 1: Comparison of 3-shot train and test accuracy@$k$ rates on GSM8K and MBPP for Gemma 2 2B checkpoints produced by various setups. We report the mean and np.std(*, ddof=1) for 3 seeds. To report train accuracy@$k$, we sample $k$ solutions to each problem at temperature 0.7 and mark it correct if any of $k$ solutions is correct. For test accuracy, we employ greedy sampling. (*): Row 2 cites the figure from the Gemma 2 report [GRP+24] which does not report sampling temperature.

## 7.3 GSM8K Results

Table 1 (first half) summarizes our results on GSM8K. We have 7000 training problems, use $k = 8$ for generate, and allocate the same total query budget of 56,000 to the labeler each round. In Appendix C, we present model responses to selected problems over the course of training.

**Baselines validate our experimental setup.** Results in the *PT* and *Gold SFT* demonstrate that: (1) our evaluation setup is in the ballpark of what is reported in the original Gemma 2 report; and (2) our fine-tuning setup indeed can yield significant improvement when the training data is human-written solutions.

**Model collapse with no curation.** In the *Do nothing* row, we recover the result from the model collapse literature that iterative fine-tuning without curation does not improve the model and leads to degraded quality.

**Comparison between curation variants.** Indeed, the present algorithm demonstrates improvements over the ReST-like variant that uses filtering only. The differences are most evident in training accuracy, which is strongly predicted by the theory. Indeed, this is in spite of the fact that *as opposed to filtering only, boosting introduces incorrect answers to the training data*. Furthermore although our theory does not address generalization, we observe that boosting results in improved test accuracy. Finally, the performance of boosting without focusing is quite close – random selection is a strong baseline – but focusing still leads to improvements, especially in terms of training accuracy.

## 7.4 MBPP Results

Table 1 (second half) summarizes our results on MBPP. We have 374 training problems, use $k = 32$ for generate, and allocate the same total query budget of 11,968 to the weak labeler in each round.

**Similar results to GSM8K for train pass rate.** In terms of train pass@$k$, we observe similar results to GSM8K experiments, that generally: *Boosting > Boosting w/o focusing > Filter only > Do nothing*. On *weaker data (B)*, *Filtering* beats *Boosting w/o focusing* in terms of pass@1.

**No clear winner for test pass rate.** While all iterative approaches outperform *Gold SFT* in terms of test pass rate, they all recover similar test performance despite differences in training accuracy. Notably, *Boosting w/o focusing* beats *Boosting*, and *Filter Only* outperforms *Boosting* with weaker data (B). One explanation is the limited amount of training data (384 examples) which prevents generalization; note that *Gold SFT* does not recover *PT* test pass rate.

# 8 Conclusion & Future Work

We have shown that under mild assumptions a modicum of curation applied to synthetic data not only avoids model collapse, but leads to arbitrarily high accuracy results. Our analysis is through the lens of boosting and, mirroring that paradigm, we define notions of *strong learners* and *weak data* to reach the theoretical conclusions. In taking this view, we provide theoretical explanations for many of the synthetic data methods used in practice.

Many interesting questions remain. An immediate avenue is further relaxing the assumptions (e.g., having nearly strong learners that only approximately match the conditional distribution) and deriving corresponding convergence rates. A broader goal is using these insights for the burgeoning field of data selection, where we must explicitly model similarities between different examples as part of the analysis.

## Author contributions

- **Sergei V** conceived the idea to analyze synthetic data training through the lens of boosting.
- **Sara B** and **Alex B** reviewed related work.
- **Everyone** developed the modeling framework.
- **Kareem A**, **Weiwei K**, **Umar S** and **Sergei V** proved the main result.
- **Alex B** designed and ran the experiments.
- **Everyone** contributed to writing the paper and framing its contributions.

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

# A Theoretical Analysis

## A.1 Proof of Theorem 5

Throughout the proof, we will write $S$ to denote a dataset where all of the labels were generated synthetically (*i.e.*, by an LLM), $D$ to denote a dataset where all of the labels were provided by `labeler`$_\beta$, and $\mathcal{D}$ to denote a dataset containing a mixture of these kinds of data. Also, only datasets denoted by $\mathcal{D}$ will contain elements whose weights can differ from 0 and 1. All other datasets will be ordinary sets.

We adopt a few simplifying assumptions and conventions. Assume that the given prompt set $P$ is non-empty. Assume that the initial LLM $g_0$ returns an incorrect response to every prompt. Assume that any LLM returns at most one correct response to any prompt. Removing the latter pair of assumptions would only speed up the convergence of Algorithm 2 to an optimal LLM, but would also further complicate its analysis. Finally, we adopt the convention that $\infty \cdot 0 = 0$. This convention is needed when Algorithm 2 constructs $\mathcal{D}_t$ via the weighted union operation, since it can happen that $\lambda_t = \infty$, but this only occurs when $D_t$ is empty.

Let $P_t^+ = \{x \in P : (x, y) \in S_t^+\}$ be the correct prompts selected by `filter`. By definition $P_t^+$ and $P_t^-$ form a partition of $P$. Furthermore, $D_t$ pairs each prompt in $P_t^-$ with the label it was assigned by `labeler`$_\beta$, and $S_t^+$ pairs each prompt in $P_t^+$ with the (synthetic) label it was assigned by the previous iteration's LLM, $g_{t-1}$. Observe that $P_t^-(x) = \sum_y D_t(x, y)$ and $P_t^+(x) = \sum_y S_t^+(x, y)$. For all $t \geq 1$ and $x \in P$ let

$$q_t(x) = \sum_y D_t(y|x) q(x, y)$$

$$q_t^+(x) = \sum_y S_t^+(y|x) q(x, y)$$

$$\bar{q}_t(x) = \sum_y \mathcal{D}_t(y|x) q(x, y)$$

be the average quality of the responses to prompt $x$ in datasets $D_t$, $S_t^+$ and $\mathcal{D}_t$, respectively. For convenience we also define $\bar{q}_0(x) = \mathrm{E}_{y \sim g_0(x)}[q(x, y)]$. Note that $\bar{q}_0(x) = 0$ for all $x \in \mathcal{X}$ by assumption.

**Lemma 6.** *For all $t \geq 1$ and $x \in P$*

$$\bar{q}_t(x) = \frac{\sum_{s=1}^t \lambda_s P_s^-(x) q_s(x) + P_s^+(x) q_s^+(x)}{\sum_{s=1}^t \lambda_s P_s^-(x) + P_s^+(x)}.$$

*Proof.* We have

$$\mathcal{D}_t(y|x) = \frac{\mathcal{D}_t(x, y)}{\sum_{y'} \mathcal{D}_t(x, y')}$$

$$= \frac{\sum_{s=1}^t \lambda_s D_s(x, y) + S_s^+(x, y)}{\sum_{y'} \sum_{s=1}^t \lambda_s D_s(x, y') + S_s^+(x, y')}$$

$$= \frac{\sum_{s=1}^t \lambda_s D_s(x, y) + S_s^+(x, y)}{\sum_{s=1}^t \lambda_s P_s^-(x) + P_s^+(x)}$$

$$= \frac{\sum_{s=1}^t \lambda_s P_s^-(x) D_s(y|x) + P_s^+(x) S_s^+(y|x)}{\sum_{s=1}^t \lambda_s P_s^-(x) + P_s^+(x)}$$

and therefore

$$\bar{q}_t(x) = \sum_y \mathcal{D}_t(y|x) q(x, y)$$

$$= \frac{\sum_y \sum_{s=1}^t \lambda_s P_s^-(x) D_s(y|x) q(x, y) + P_s^+(x) S_s^+(y|x) q(x, y)}{\sum_{s=1}^t \lambda_s P_s^-(x) + P_s^+(x)}$$

$$= \frac{\sum_{s=1}^{t} \lambda_s P_s^-(x) q_s(x) + P_s^+(x) q_s^+(x)}{\sum_{s=1}^{t} \lambda_s P_s^-(x) + P_s^+(x)} \qquad \square$$

**Lemma 7.** *For all $t \geq 1$ and $x \in P$ we have $\bar{q}_t(x) = 0$ if and only if $\bar{q}_{t-1}(x) = 0$ and $q_t(x) = 0$.*

*Proof.* Suppose $\bar{q}_t(x) = 0$. By Lemma 6 this implies $\lambda_s P_s^-(x) q_s(x) + P_s^+(x) q_s^+(x) = 0$ for $s \in \{1, \ldots, t\}$, and therefore $\bar{q}_{t-1}(x) = 0$. This implies that $x$ cannot be correctly labeled in $S_t$, and therefore $P_t^-(x) = 1$. Since $\alpha > 0$ we have $\lambda_t > 0$. And since $\lambda_t P_t^-(x) q_t(x) = 0$ we must have $q_t(x) = 0$.

Now suppose $\bar{q}_{t-1}(x) = 0$ and $q_t(x) = 0$. Since $\bar{q}_{t-1}(x) = 0$ then again by Lemma 6 we have $\lambda_s P_s^-(x) q_s(x) + P_s^+(x) q_s^+(x) = 0$ for $s \in \{1, \ldots, t-1\}$. The fact that $\bar{q}_{t-1}(x) = 0$ also implies that $x$ cannot be correctly labeled in $S_t$, and therefore $P_t^+(x) = 0$. And since $q_t(x) = 0$ we have $\lambda_t P_t^-(x) q_t(x) + P_t^+(x) q_t^+(x) = 0$, which implies $\bar{q}_t(x) = 0$. $\qquad \square$

**Lemma 8.** *Let $t \geq 1$ and $x \in P$. If $x \in P_t^-$ then $q_t(x) \in \{0, 1\}$. If $x \in P_t^+$ then $q_t^+(x) = 1$.*

*Proof.* Note that $D_t$ contains each prompt only once (by Definition 3 of $\texttt{labeler}_\beta$), and $S_t^+$ contains only correctly labeled prompts (by Definition 2 of $\texttt{filter}$). The lemma follows from the definitions of $P_t^-$, $P_t^+$, $q_t(x)$ and $q_t^+(x)$. $\qquad \square$

**Lemma 9.** *If $a, b, c, d \geq 0$ satisfy $a \leq b$, $c \geq d$ and $b > 0$ then*

$$\frac{a+c}{b+c} \geq \frac{a+d}{b+d}.$$

*Proof.* If $c = d$ then clearly the lemma holds with equality. Otherwise if $c > d$ then

$$
\begin{aligned}
& \frac{a+c}{b+c} \geq \frac{a+d}{b+d} & \\
\Leftrightarrow\ & (a+c)(b+d) \geq (a+d)(b+c) & b > 0 \\
\Leftrightarrow\ & ab + bc + ad + cd \geq ab + bd + ac + cd & \\
\Leftrightarrow\ & bc + ad \geq bd + ac & \\
\Leftrightarrow\ & b(c-d) \geq a(c-d) & \\
\Leftrightarrow\ & b \geq a & c > d \qquad \square
\end{aligned}
$$

Our analysis relies on conditioning on the fact that once the quality of a particular prompt, $x$, is high enough, it is always selected by $\texttt{filter}$ and is never sent to $\texttt{labeler}_\beta$. Formally, fix the number of iterations, $T$, the set of prompts, $P$, and the quality of the weak data, $\beta$. We define event $E$, as follows:

$$\text{Event } E \equiv \text{For all } t \in [T] \text{ and } x \in P \text{ if } \bar{q}_{t-1}(x) \geq \beta \text{ then } x \notin P_t^-,$$

and to simplify notation, we drop the dependence of $E$ on $T$, $P$ and $\beta$.

**Lemma 10.** *If the repeat parameter $k \geq \frac{2 \log T + \log |P|}{\beta \gamma}$ then event $E$ occurs with probability at least $1 - \frac{1}{T}$.*

*Proof.* By the definition of $\texttt{generate}$ (Definition 4), each $x \in P$ is labeled $k$ times by $g_{t-1}$ in iteration $t$, with each label drawn independently from distribution $g_{t-1}(x)$. Thus we know that if $\bar{q}_{t-1}(x) \geq \beta$ then $x \in P_t^-$ with probability at most $(1 - \beta\gamma)^k$. This is because the probability that a synthetic label does not prevent a prompt from being added to $P_t^-$ is at most $1 - \beta + \beta(1-\gamma) = 1 - \beta\gamma$, which is the probability that the label is low-quality *or* that it is high-quality but is not recognized by $\texttt{filter}_\gamma$ (Definition 2). Therefore

$$\Pr[\neg E] = \Pr\left[\exists t \in [T] \text{ and } x \in P \text{ such that } \bar{q}_{t-1}(x) \geq \beta \text{ and } x \in P_t^-\right]$$

$$\leq \sum_{t=1}^{T} \sum_{x \in P} \Pr[\bar{q}_{t-1}(x) \geq \beta \text{ and } x \in P_t^-]$$

$$\leq T|P|(1-\beta\gamma)^k$$
$$\leq T|P|\exp(-\beta\gamma k)$$
$$\leq T|P|\exp(-2\log T - \log|P|) \qquad\qquad \text{Assumption about } k$$
$$= T|P|\frac{1}{T^2}\frac{1}{|P|}$$
$$= \frac{1}{T} \qquad\qquad\qquad\qquad\qquad\qquad\qquad\qquad\qquad \Box$$

The next result is our key lemma. It says that if event $E$ occurs then (a) $P_t^-$ contains all and only the prompts that must have been incorrectly labeled by the previous iteration's LLM, (b) the size of $P_t^-$ shrinks exponentially over time, (c) once a prompt is outside $P_t^-$ it remains that way, and (d) prompts outside of $P_t^-$ are correctly labeled by the previous iteration's LLM with a probability that is bounded above zero.

**Lemma 11.** *Fix $T$. Let $1 \le t \le T$ and $x \in P$. If event $E$ occurs then all of the following hold:*

*(a) $x \in P_t^-$ if and only if $\bar{q}_{t-1}(x) = 0$.*

*(b) $|P_r^-| \le (1-\beta)^{r-s}|P_s^-|$ for all $r, s \in [t]$ such that $r \ge s$.*

*(c) There exists $r \in [t]$ such that $x \in P_s^-$ for all $s \in [r]$ and $x \notin P_s^-$ for all $s \in [t] \setminus [r]$.*

*(d) Let $r \in [t]$ satisfy the conditions of part (c). If $r < t$ then*

$$\bar{q}_t(x) \ge \frac{\alpha + t - r}{\frac{\alpha(1-(1-\beta)^r)}{\beta} + t - r} \ge \beta.$$

*Proof.* The proof will proceed by induction. We begin by proving the base case, $t = 1$. To prove part (a), note that by assumption we have $\bar{q}_0(x) = \mathrm{E}_{y \sim g_0(x)}[q(x, y)] = 0$, so we only need to show that $x \in P_1^-$. Since $\mathrm{E}_{y \sim g_0(x)}[q(x, y)] = 0$, we know that $x$ cannot be correctly labeled in $S_1$, which implies $x \in P_1^-$. Part (b) follows immediately from the observation that when $t = 1$ we have $r = s = 1$. Part (c) holds immediately by letting $r = 1$, since in this case $[t] \setminus [r]$ is empty, and we have already shown $x \in P_1^-$ in part (a). Part (d) holds vacuously because $r < t$ must be false when $t = 1$.

Now assume for induction that the lemma holds for a fixed $t \ge 1$. We will prove the lemma for the case $t + 1$. To prove part (a), first assume $\bar{q}_t(x) = 0$, which is the premise of the 'if' direction. By Definition 1 we have

$$\mathrm{E}_{y \sim g_t(x)}[q(x, y)] = \sum_y \mathcal{D}_t(x, y)q(x, y) = \bar{q}_t(x) = 0$$

which implies that $x$ cannot be correctly labeled in $S_{t+1}$, and therefore $x \in P_{t+1}^-$. Now assume $x \in P_{t+1}^-$, which is the premise of the 'only if' direction. To force a contradiction, assume that $\bar{q}_t(x) > 0$. By part (d) of the inductive hypothesis, this implies $\bar{q}_t(x) \ge \beta$. Since event $E$ occurred, we have that $x \notin P_{t+1}^-$, which is a contradiction. This completes the proof of part (a).

To prove part (b), choose any $r, s \in [t+1]$ such that $r \ge s$. If $r = s$, part (b) follows immediately. If $r < t+1$ and $s < t+1$ then part (b) follows from the inductive hypothesis. Henceforth assume $s < r = t+1$. Let

$$D_t^+ = \{(x, y) \in D_t : q(x, y) = 1\}$$

be the subset of $D_t$ that is correctly labeled. We have

$$(1-\beta)|P_t^-| = (1-\beta)|D_t|$$
$$\ge |D_t| - |D_t^+| \qquad\qquad \text{Definition 3 of } \texttt{labeler}_\beta$$
$$= \sum_{x,y} D_t(x, y) - \sum_{x,y} D_t(x, y)q(x, y)$$
$$= \sum_{x,y} D_t(x, y)(1 - q(x, y))$$

$$= \sum_{x,y} P_t^-(x)D_t(y|x)(1 - q(x,y))$$

$$= \sum_x P_t^-(x)(1 - q_t(x))$$

$$= \sum_x P(x)\mathbf{1}\{\bar{q}_{t-1}(x) = 0\}(1 - q_t(x)) \qquad \text{Inductive hypothesis, part (a)}$$

$$= \sum_x P(x)\mathbf{1}\{\bar{q}_{t-1}(x) = 0\}\mathbf{1}\{q_t(x) = 0\} \qquad \text{Lemma 8}$$

$$= \sum_x P(x)\mathbf{1}\{\bar{q}_t(x) = 0\} \qquad \text{Lemma 7}$$

$$= \sum_x P_{t+1}^-(x) \qquad \text{Part (a)}$$

$$= |P_{t+1}^-|$$

and therefore

$$|P_{t+1}^-| \le (1 - \beta)|P_t^-| \le (1 - \beta)(1 - \beta)^{t-s}|P_s^-| = (1 - \beta)^{t+1-s}|P_s^-| = (1 - \beta)^{r-s}|P_s^-|$$

where the second inequality follows from the inductive hypothesis. This completes the proof of part (b).

To prove part (c), we must prove the existence of a satisfying iteration $r \in [t + 1]$. Let $r' \in [t]$ be the iteration that satisfies part (c) of the inductive hypothesis. If $r' = t$ and $x \notin P_{t+1}^-$ then we can let $r = t$. If $r' = t$ and $x \in P_{t+1}^-$ then we can let $r = t + 1$. If $r' < t$ then we only have to show $x \notin P_{t+1}^-$, because in that case we can let $r = r'$. Since $r' < t$ we have $x \notin P_t^-$, and by part (a) we have $\bar{q}_{t-1}(x) > 0$. By Lemma 6 we have $\bar{q}_t(x) > 0$, and thus by part (a) again we have $x \notin P_{t+1}^-$. This concludes the proof of part (c).

To prove part (d), let $r \in [t + 1]$ be the satisfying iteration from part (c). Note that $r < t + 1$ by the premise of part (d). We first prove that

$$P_s^-(x)q_s(x) = 0 \text{ for all } s \in [r - 1]. \tag{2}$$

Suppose for contradiction that Eq. (2) is not true, which implies that $P_s^-(x)q_s(x) > 0$ for some $s \in [r - 1]$. By Lemma 6 and the fact that $\alpha > 0$ we have $\bar{q}_s(x) > 0$, which implies by part (a) that $x \notin P_{s+1}^-$, which contradicts part (c). Thus we have proved Eq. (2). We next prove that

$$P_r^-(x)q_r(x) = 1. \tag{3}$$

Suppose for contradiction that Eq. (3) is not true, which implies by part (c) and Lemma 8 that $P_r^-(x)q_r(x) = 0$. Thus by Eq. (2) we have $P_s^-(x)q_s(x) = 0$ for $s \in [r]$. We also have by part (c) that $P_s^+(x) = 0$ for $s \in [r]$. Thus by Lemma 6 we have $\bar{q}_r(x) = 0$, and this implies by part (a) that $x \in P_{r+1}^-$, which by $r < t + 1$ contradicts part (c). Thus we have proved Eq. (3). We are now ready to complete the proof of part (d). We have

$$\bar{q}_{t+1}(x) = \frac{\sum_{s=1}^{t+1} \lambda_s P_s^-(x)q_s(x) + P_s^+(x)q_s^+(x)}{\sum_{s=1}^{t+1} \lambda_s P_s^-(x) + P_s^+(x)} \qquad \text{Lemma 6}$$

$$= \frac{\sum_{s=1}^{t+1} \frac{\alpha}{|D_s|} P_s^-(x)q_s(x) + P_s^+(x)q_s^+(x)}{\sum_{s=1}^{t+1} \frac{\alpha}{|D_s|} P_s^-(x) + P_s^+(x)}$$

$$= \frac{\frac{\alpha}{|D_r|} + \sum_{t=r+1}^{t+1} 1}{\sum_{s=1}^{r} \frac{\alpha}{|D_s|} + \sum_{t=r+1}^{t+1} 1} \qquad \text{Part (c), Lemma 8, Eq. (2) and Eq. (3)}$$

$$= \frac{\frac{\alpha}{|D_r|} + t - r + 1}{\sum_{s=1}^{r} \frac{\alpha}{|D_s|} + t - r + 1}$$

$$= \frac{\frac{\alpha}{|D_r|} + t - r + 1}{\sum_{s=1}^{r} \frac{\alpha}{|D_s|} + t - r + 1}$$

$$= \frac{\frac{\alpha}{|P_r^-|} + t - r + 1}{\sum_{s=1}^{r} \frac{\alpha}{|P_s^-|} + t - r + 1} \qquad \text{Definitions of } P_t^- \text{ and } P_t^+$$

$$\geq \frac{\frac{\alpha}{|P_r^-|} + t - r + 1}{\frac{\alpha}{|P_r^-|} \sum_{s=1}^{r} (1-\beta)^{r-s} + t - r + 1} \qquad \text{Part (b)}$$

$$= \frac{\frac{\alpha}{|P_r^-|} + t - r + 1}{\frac{\alpha}{|P_r^-|} \sum_{s=0}^{r-1} (1-\beta)^s + t - r + 1}$$

$$= \frac{\frac{\alpha}{|P_r^-|} + t - r + 1}{\frac{\alpha(1-(1-\beta)^r)}{\beta|P_r^-|} + t - r + 1} \qquad \text{Geometric series formula}$$

$$= \frac{\alpha + |P_r^-|(t - r + 1)}{\frac{\alpha(1-(1-\beta)^r)}{\beta} + |P_r^-|(t - r + 1)}$$

$$\geq \frac{\alpha + t - r + 1}{\frac{\alpha(1-(1-\beta)^r)}{\beta} + t - r + 1} \qquad \text{Lemma 9 and } |P_r^-| \geq 1 \text{ (by choice of } r)$$

which proves the first inequality of part (d). Continuing from above

$$\bar{q}_{t+1}(x) \geq \frac{\alpha + t - r + 1}{\frac{\alpha(1-(1-\beta)^r)}{\beta} + t - r + 1} \qquad \text{From above}$$

$$\geq \frac{\alpha}{\frac{\alpha(1-(1-\beta)^r)}{\beta}} \qquad \text{Lemma 9}$$

$$= \frac{\beta}{1 - (1-\beta)^r} \qquad \alpha > 0$$

$$\geq \beta \qquad \beta > 0$$

which proves the second inequality of part (d). $\qquad\square$

We are now ready to complete the proof of Theorem 5. Assume that event $E$ occurs, which by Lemma 10 happens with probability at least $1 - \frac{1}{T}$. For each prompt $x \in P$ let $r_x$ be the iteration that satisfies Lemma 11(c) when the lemma is applied to prompt $x$ and iteration $T$. Let $r = \frac{\log(2/\varepsilon)}{\beta}$, and note that by assumption $r < T$. We have

$$\Pr_{x \sim P, y \sim g_T(x)}[q(x, y) = 1]$$

$$= \mathrm{E}_{x \sim P, y \sim g_T(x)}[q(x, y)]$$

$$= \mathrm{E}_{x \sim P}\left[ \sum_y \mathcal{D}_T(y|x) q(x, y) \right] \qquad \text{Definition 1}$$

$$= \mathrm{E}_{x \sim P}[\bar{q}_T(x)]$$

$$\geq \mathrm{E}_{x \sim P}[\bar{q}_T(x) \mid r_x \leq r] \Pr_{x \sim P}[r_x \leq r]$$

$$= \mathrm{E}_{x \sim P}[\bar{q}_T(x) \mid r_x \leq r] \Pr_{x \sim P}[x \notin P_{r+1}^-] \qquad \text{Lemma 11(c)}$$

$$= \mathrm{E}_{x \sim P}[\bar{q}_T(x) \mid r_x \leq r] \left( 1 - \frac{|P_{r+1}^-|}{|P|} \right)$$

$$= \mathrm{E}_{x \sim P}[\bar{q}_T(x) \mid r_x \leq r] \left( 1 - \frac{|P_{r+1}^-|}{|P_1^-|} \right) \qquad \text{Lemma 11(a)}$$

$$\geq \mathrm{E}_{x \sim P}[\bar{q}_T(x) \mid r_x \leq r] \left( 1 - (1-\beta)^r \right) \qquad \text{Lemma 11(b)}$$

$$\geq \min_{x : r_x \leq r} \frac{\alpha + T - r_x}{\frac{\alpha(1-(1-\beta)^{r_x})}{\beta} + T - r_x} \left( 1 - (1-\beta)^r \right) \qquad \text{Lemma 11(d)}$$

$$\geq \min_{x : r_x \leq r} \frac{\alpha + T - r_x}{\frac{\alpha}{\beta} + T - r_x} \left( 1 - (1-\beta)^r \right)$$

$$\geq \frac{\alpha + T - r}{\frac{\alpha}{\beta} + T - r} \left(1 - (1 - \beta)^r\right) \hspace{4cm} \text{Lemma 9}$$

$$\geq \frac{\alpha + T - r}{\frac{\alpha}{\beta} + T - r} \left(1 - e^{-\beta r}\right)$$

$$= \frac{\alpha + T - r}{\frac{\alpha}{\beta} + T - r} \left(1 - \frac{\varepsilon}{2}\right)$$

Since

$$T \geq \frac{\log(2/\varepsilon)}{\beta} + \frac{2\alpha}{\beta\varepsilon} = r + \frac{2\alpha}{\beta\varepsilon}$$

it is easy to show via algebra that

$$\frac{\alpha + T - r}{\frac{\alpha}{\beta} + T - r} \geq 1 - \frac{\varepsilon}{2}$$

and plugging this into the final expression above proves $\Pr_{x \sim P, y \sim g_T(x)}[q(x, y) = 1] \geq \left(1 - \frac{\varepsilon}{2}\right)^2 \geq 1 - \varepsilon$, which proves the theorem.

# B   Additional Plots

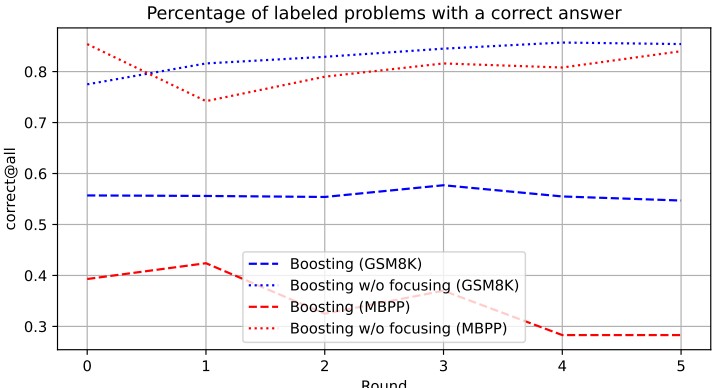

Figure 3: Labeler accuracy across rounds. These results use *weak data (A)*. Since training accuracy increases across rounds, the weak labeler gets more queries per question in both cases. Despite this, for *Boosting* we see that accuracy is relatively constant for GSM8K and decreasing for MBPP. This is because we focus on increasingly harder problems. In *Boosting w/o focusing*, we observe labeler accuracy increasing because we do not focus labeler efforts on the highest difficulty problems.

In Figure 4, we plot average response length on the test set as a function of training rounds. Average response length does not change significantly throughout training, and incorrect answers are longer on average.

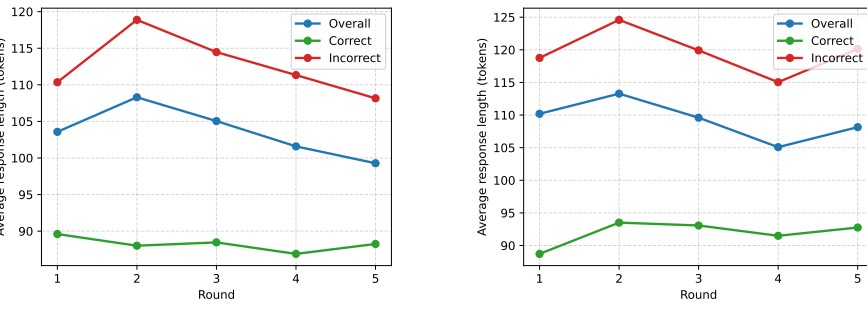

(a) Average response length, *Boosting (A)*       (b) Average response length, *Boosting (B)*

Figure 4: Average length of responses to GSM8K test set problems across rounds for *Boosting* experiments.

## C  Examples

In Tables 3 and 4 we present specific examples of the model improving its answer to a particular question across rounds.

| *Question* | Bill starts on the 3rd floor. He rides the elevator up to the floor that's equal to 4 times his starting floor plus 6. What floor is Bill on now? |
|---|---|
| *Response (round 1)* | Let's use algebra to solve this problem. Let x be the number of floors Bill is on now. 
 The number of floors Bill is on now is equal to 4x + 6. 
 We know that Bill starts on the 3rd floor, so x = 3. 
 Therefore, Bill is on the 3rd floor now. 
 The answer is 3. ✗ |
| *Response (round 2)* | Let's use algebra to solve this problem. Let x be the floor Bill is on now. 
 The floor Bill is on now is equal to 4x + 6. 
 We know that Bill starts on the 3rd floor, so x = 3. 
 Therefore, Bill is on the 3rd floor now. 
 The answer is 3. ✗ |
| *Response (round 3)* | Let us solve it using algebra. Let x be the floor Bill is on now. 
 The floor he rides up to is 4x + 6. 
 Therefore, the floor he rides up to is equal to the floor he starts on plus 6. 
 So, we have the equation: 
 x = 3 + 6 
 x = 9 
 Therefore, Bill is on the 9th floor now. 
 The answer is 9. ✗ |
| *Response (round 4)* | Let us solve it using algebra. Let x be the floor Bill is on now. 
 The floor he is on now is 4x + 6. 
 We know that he starts on the 3rd floor, so x = 3. 
 Therefore, the floor he is on now is 4(3) + 6 = 12 + 6 = 18. 
 Bill is on the 18th floor now. 
 The answer is 18. ✓ |
| *Response (round 5)* | Let us solve it using algebra. Let x be the floor Bill is on now. 
 The floor he is on now is 4x + 6. 
 We know that he starts on the 3rd floor, so x = 3. 
 Therefore, the floor he is on now is 4(3) + 6 = 12 + 6 = 18. 
 Bill is on the 18th floor now. 
 The answer is 18. ✓ |

Table 3: Tracking responses to problem `id=253` from the GSM8K test set, across rounds of a *Boosting (A)* run. The model always sets up the right expression ($4x + 6$), but in the first two rounds reports $x = 3$ as the answer instead of substituting it into the expression. The correct substitution is done in round 4, and the correct answer is maintained in round 5.

| | |
|---|---|
| *Question* | The Kennel house keeps 3 German Shepherds and 2 Bulldogs. If a German Shepherd consumes 5 kilograms of dog food and a bulldog consumes 3 kilograms of dog food per day. How many kilograms of dog food will they need in a week? |
| *Response (round 1)* | Let's solve this problem using algebra.
**Step 1: Find the total amount of dog food consumed by the German Shepherds in a week:**
* German Shepherds: 3 dogs
* Dog food consumed per day: 5 kilograms
* Dog food consumed in a week: 3 dogs * 5 kilograms/dog = 15 kilograms
**Step 2: Find the total amount of dog food consumed by the Bulldogs in a week:**
* Bulldogs: 2 dogs
* Dog food consumed per day: 3 kilograms
* Dog food consumed in a week: 2 dogs * 3 kilograms/dog = 6 kilograms
**Step 3: Find the total amount of dog food consumed in a week:**
* Total dog food consumed: 15 kilograms + 6 kilograms = 21 kilograms
Therefore, the kennel house will need 21 kilograms of dog food in a week.
The answer is 21. ✗ |
| *Response (round 2)* | Let's calculate the total amount of dog food needed in a week.
**German Shepherds:**
- Number of German Shepherds: 3
- Dog food consumption per day: 5 kg
- Dog food consumption in a week: 3 * 7 = 21 kg
**Bulldogs:**
- Number of Bulldogs: 2
- Dog food consumption per day: 3 kg
- Dog food consumption in a week: 2 * 7 = 14 kg
**Total dog food consumption in a week:**
- Total dog food consumption in a week: 21 kg + 14 kg = 35 kg
Therefore, the kennel house will need 35 kilograms of dog food in a week.
The answer is 35. ✗ |
| *Response (round 3)* | Let's solve this problem using algebra.
**German Shepherds:**
* Number of German Shepherds: 3
* Dog food consumed per day: 5 kilograms
* Dog food consumed in a week: 3 * 7 = 21 kilograms
**Bulldogs:**
* Number of Bulldogs: 2
* Dog food consumed per day: 3 kilograms
* Dog food consumed in a week: 2 * 7 = 14 kilograms
**Total dog food consumed in a week:**
* Total dog food consumed in a week: 21 kilograms + 14 kilograms = 35 kilograms
Therefore, the Kennel house will need 35 kilograms of dog food in a week.
The answer is 35. ✗ |
| *Response (round 4)* | Let's calculate the total dog food needed per day:
* German Shepherds: 3 * 5 = 15 kilograms
* Bulldogs: 2 * 3 = 6 kilograms
* Total dog food needed per day: 15 + 6 = 21 kilograms
Now, let's calculate the total dog food needed in a week:
* 7 days * 21 kilograms/day = 149 kilograms
Therefore, the kennel house will need 149 kilograms of dog food in a week.
The answer is 149. ✗ |
| *Response (round 5)* | Let's calculate the total dog food consumption per day:
* German Shepherds: 3 * 5 = 15 kilograms
* Bulldogs: 2 * 3 = 6 kilograms
* Total: 15 + 6 = 21 kilograms
Now, let's calculate the total dog food consumption in a week:
* 7 days * 21 kilograms/day = 147 kilograms
Therefore, the Kennel house will need 147 kilograms of dog food in a week.
The answer is 147. ✓ |

Table 4: Tracking responses to problem `id=1266` from the GSM8K test set, across rounds of a *Boosting (B)* run. The round 1 response gets the correct amount per day (21), but does not multiply to get the correct value for the week. Rounds 2 and 3 fail to incorporate the amount of food eaten per dog. Round 4 gets the correct approach, but fails the final multiplication step, which is corrected in the final round.

# D  Prompts

## D.1  GSM8K

As an expert problem solver solve step by step the following mathematical questions.

Q:
Tina makes $18.00 an hour. If she works more than 8 hours per shift, she is eligible for overtime, which is paid by your hourly wage + 1/2 your hourly wage. If she works 10 hours every day for 5 days, how much money does she make?

A:
Here's how to calculate Tina's earnings:

**Regular Time:**

- Hours per shift: 8 hours

- Wage per hour: $18.00

- Regular pay per shift: 8 hours * $18.00/hour = $144.00

**Overtime:**

- Overtime hours per shift: 10 hours - 8 hours = 2 hours

- Overtime pay per hour: $18.00 + ($18.00 / 2) = $27.00

- Overtime pay per shift: 2 hours * $27.00/hour = $54.00

**Total per day:**

- Regular pay + overtime pay: $144.00/shift + $54.00/shift = $198.00/day

**Total for 5 days:**

- 5 days * $198.00/day = $990.00

**Therefore, Tina will make $990.00 in 5 days.**

The answer is 990.

Q:
Abigail is trying a new recipe for a cold drink. It uses 1/4 of a cup of iced tea and 1 and 1/4 of a cup of lemonade to make one drink. If she fills a pitcher with 18 total cups of this drink, how many cups of lemonade are in the pitcher?

A:
## Ambiguity in the Problem Statement:

There is one main ambiguity in the problem statement:

**Total volume vs. Number of servings:** The statement "18 total cups of this drink" could be interpreted in two ways:

* 18 cups of the combined volume

* 18 individual servings

Let us assume the interpretation "18 cups of the combined volume".

## Solution assuming 18 cups of combined volume:

**Step 1: Find the proportion of lemonade in one drink:**

- Lemonade: 1 1/4 cups

- Iced tea: 1/4 cup

- Total: 1 1/4 + 1/4 = 1 1/2 cups

- Lemonade proportion: (1 1/4) / (1 1/2) = 5/6

**Step 2: Calculate the amount of lemonade in the pitcher:**

- Total volume: 18 cups

- Lemonade proportion: 5/6

- Volume of lemonade: 18 * (5/6) = 15 cups

Therefore, there are 15 cups of lemonade in the pitcher.

The answer is 15.

Q:
A deep-sea monster rises from the waters once every hundred years to feast on a ship and sate its hunger. Over three hundred years, it has consumed 847 people. Ships have been built larger over time, so each new ship has twice as many people as the last ship. How many people were on the ship the monster ate in the first hundred years?

A:
Let us solve it using algebra. Let x be the number of people on the ship the monster ate in the first hundred years.

The number of people on the ship eaten in the second hundred years is 2x, and in the third hundred years is 4x.

Therefore, the total number of people eaten over three hundred years is x + 2x + 4x = 847.

Combining like terms, we get 7x = 847.

Dividing both sides by 7, we find x = 121.

Therefore, there were 121 people on the ship the monster ate in the first hundred years.

The answer is 121.

Q:
{problem}

A:

## D.2 MBPP

You are an expert Python programmer. Solve the following Python programming problems.

Q:
Write a function to find the similar elements from the given two tuple lists.
Your code should pass these tests:
```
assert similar_elements((3, 4, 5, 6),(5, 7, 4, 10)) == (4, 5)
assert similar_elements((1, 2, 3, 4),(5, 4, 3, 7)) == (3, 4)
assert similar_elements((11, 12, 14, 13),(17, 15, 14, 13)) == (13, 14)
```

A:

```
def similar_elements(test_tup1, test_tup2):
  res = tuple(set(test_tup1) & set(test_tup2))
  return (res)
```

Q:
Write a python function to identify non-prime numbers.
Your code should pass these tests:

```
assert is_not_prime(2) == False
assert is_not_prime(10) == True
assert is_not_prime(35) == True
```

A:

```
import math
def is_not_prime(n):
  result = False
  for i in range(2,int(math.sqrt(n)) + 1):
  if n % i == 0:
    result = True
  return result
```

Q:
Write a function to find the largest integers from a given list of numbers using heap queue algorithm.
Your code should pass these tests:

```
assert heap_queue_largest( [25, 35, 22, 85, 14, 65, 75, 22, 58],3)==[85, 75, 65]
assert heap_queue_largest( [25, 35, 22, 85, 14, 65, 75, 22, 58],2)==[85, 75]
assert heap_queue_largest( [25, 35, 22, 85, 14, 65, 75, 22, 58],5)==[85, 75, 65, 58, 35]
```

A:

```
import heapq as hq
def heap_queue_largest(nums,n):
  largest_nums = hq.nlargest(n, nums)
  return largest_nums
```

Q:
{problem}
Your code should past these tests:
{tests}

A:

# E   Additional Discussion

**Weighting Examples**   Theorem 5 requires $\alpha > 0$ and $\beta > 0$, and it is easy to see that both assumptions are necessary. If $\alpha = 0$ then each training set $\mathcal{D}_t$ contains no exogenously labeled data, and if $\beta = 0$ then all of the exogenously labeled data could be incorrect. In either case, and given a worst-case initial LLM $g_0$ that returns an incorrect response to every prompt, the learner has no way to determine the correct response to any prompt.

The astute reader will note that $\alpha$ can also be set arbitrarily close to zero in Theorem 5 without weakening the convergence guarantee. This curious property is a consequence of an idealized assumption that we made for the sake of analytical tractability. Specifically, we assumed that the learner can match the conditional response distribution of every prompt in the training data, no matter how infrequently the prompt appears in the data (see Definition 1). In practice, constraints on training time and model size will prevent a learner from perfectly fitting the training data. So it would be useful to extend our results to account for the possibility of an imperfect learner, and we expect that any such extension would imply a non-zero lower bound on $\alpha$. Nonetheless, our current results tell us something interesting – computational limitations are the *only* barrier to learning an arbitrarily good LLM, and not, as one might expect, the quality of the weak labeler.

**Filtering Non-Synthetic Data**   Algorithm 2 has the property that it only applies filtering on LLM-generated data. As discussed, this accurately models existing methods in the literature.

However, if we consider applying the quality function $q$ on data produced by the weak labeler (that is, data that is not LLM-generated), then there is an alternate solution to the data generation problem. It can be shown that $O(\log(1/\varepsilon)/\beta)$ invocations of the weak labeler would suffice to correctly label all but $\varepsilon$ fraction of the prompts in $P$, and such a dataset could be given to a strong learner to produce an LLM that achieves $O(\varepsilon)$ error. It is worthwhile to reason about why such a solution cannot be deployed in practice.

First, and most crucially – it has been repeatedly demonstrated empirically in the literature that training directly on 100% correct human responses is sub-optimal compared to training on self-generated synthetic data [SCA$^+$24, ZWMG22]. Indeed this is confirmed in our own experiments where we find that *finetuning on correct answers (Gold SFT) is dramatically outperformed by recursive training*, hence motivating our theoretical study of the problem.

The weak data assumption specifies that $\beta$, while arbitrary, is bounded away from zero. Just as the weak learning assumption might not hold in classical boosting, the weak data assumption might not hold in our setting. We argue that iteratively filtering the weak labeler's output should result in a precipitous drop in the fraction of correctly-labeled examples. As an example, suppose human labelers provide good responses to the $\beta_1$ easiest coding prompts in some prompt set. One should expect that asking similarly-qualified labelers to respond to the remaining prompts results in a $\beta_2 \ll \beta_1$ yield of quality responses, as all but the easiest prompts have been answered. In contrast, a continually improved LLM endows a human with more flexibility for future responses, such as rewriting nearly high-quality solutions provided by the last iteration of LLM, making a non-vanishing $\beta$ a much more reasonable assumption.

While it keeps the setting simple to presume that $q$ can be evaluated on any labeled example, this is an overly permissive assumption. LLM-generated synthetic data can be made to include reasoning traces, and often produces responses that the LLM itself can verify as high quality. This facilitates the construction of automated quality checkers, which are much more difficult to construct when the labels are produced by a human, and therefore contain reasoning traces and responses that are unfamiliar to the current generation of LLM. This is born out in the literature, where quality verification of LLM-generated synthetic data is relatively easy to implement [SCA$^+$24, YPF$^+$24, ZWMG22].

Finally, and somewhat remarkably, Algorithm 2 with $\alpha = \epsilon$ achieves the same finite-time error rate as this baseline while only ever evaluating the quality of LLM-generated data. Thus, the approach taken in practice matches the convergence rate that would be experienced under a much more powerful set of assumptions.

## F  Additional Experimental Results

In this section, we provide additional experimental results. Specifically, we conduct the following experiments:

- **Off-policy boosting:** We fixed the learner to be Gemma 2 2B and switched out the labeler from Gemma 2 2B (on-policy) to either Gemma 1 2B or Gemma 7B. The stronger off-policy Gemma 7B model outperforms the on-policy Gemma 2 2B model. Results are pictured in Figure 5. The weaker Gemma 1 2B significantly underperforms on-policy Gemma 2 2B, but after 5 rounds roughly approaches the performance of *filter only*. Note that the Gemma 1 2B model achieves only 11% on GSM8K, and based on our theory, we should expect to require more iterations with a weaker labeler.

- **Running longer and observing improvement plateau:** In several prior works, it was reported that improvement plateaus after a few rounds. We run some experiments longer ($>5$ rounds). Indeed, in Figure 6 we see limited improvements in test accuracy ($\approx 2\%$) from additional rounds, despite persistent improvements in training accuracy.

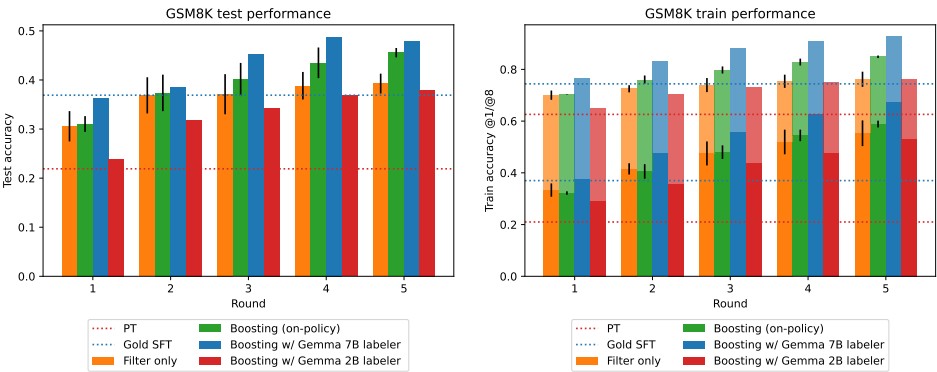

Figure 5: We experiment with *off-policy labelers* on GSM8K, plotting performance across rounds. *Boosting (on-policy)* is the setting in all prior experiments, employing Gemma 2 2B PT as the labeler. We see improvement from using the stronger Gemma 7B as our labeler. The weaker Gemma 1 2B performs much worse, but approaches the results from *Filter only* after 5 rounds.

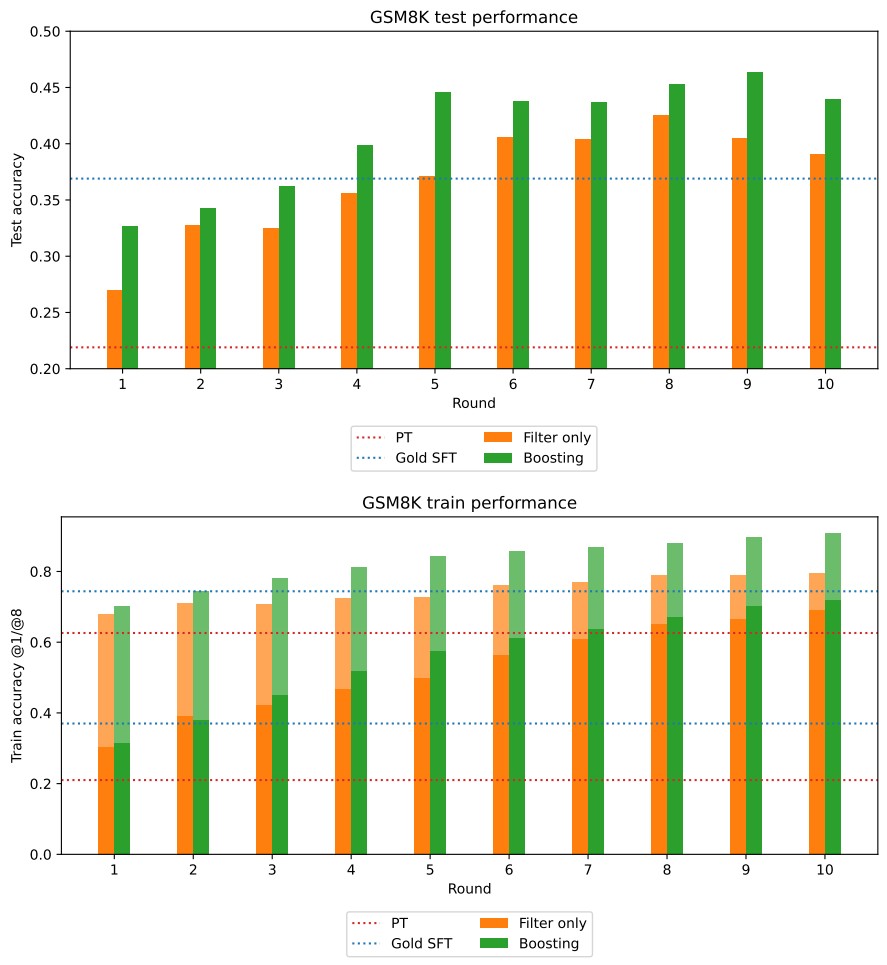

Figure 6: We plot test and train performance of *Filter only* and *Boosting* on GSM8K across more rounds (10). We see limited improvements in test accuracy (≈2%) despite persistent improvements in train accuracy.

