# OpenReview forum: "Escaping Collapse: The Strength of Weak Data for Large Language Model Training"
_NeurIPS.cc/2025/Conference — NeurIPS 2025 poster_

### Official Review · Reviewer_BnJF · 2025-06-24

**Clarity:** 4
**Significance:** 3
**Originality:** 3
**Rating:** 5
**Confidence:** 5

**Summary:**

In this paper authors introduce a theoretical framework on understanding an important phenomenon - model collapse - as well as ways of escaping it. The authors connect the ideas of weak data and strong learner with a classic ML algorithm - boosting. Experiments validate theoretical results.

**Questions:**

For the filter-only setup, is the model trained on less samples (than w/ boosting) since we filter the bad ones out?

**Ethical Concerns:**

["NO or VERY MINOR ethics concerns only"]

**Final Justification:**

Thank you for providing additional clarifications. I will hold to the rating of 5.

**Limitations:**

yes

**Paper Formatting Concerns:**

Citations throughout the paper do not follow the guidelines.

**Quality:**

4

**Strengths And Weaknesses:**

### Strengths

- Well-written and solid theoretical grounds.
- Fully-accessible model and datasets which makes it fully reproducible.
- Experimental results are in line with the theoretical analysis.
- Two datasets - GSM8k (math) and MBPP (coding).

### Weaknesses

- When T increases it seems from the experimental data (Appendix F) after T>5 improvements are diminishing. No explanation or analysis is given.
- Only one model (Gemma2 2B) and only one prompting method (when fine-tuning) - 3-shot prompting.

---

> ### Author Rebuttal · Authors · 2025-07-31
>
> We thank the reviewer for their positive assessment of our paper. We appreciate their recognition that (paraphrasing from their response) our paper is well-written, theoretically solid, and contains experiments that confirm the theoretical analysis.
>
> We respond to their concerns below.
>
> **Weakness:** *Only one model (Gemma2 2B) and only one prompting method (when fine-tuning) - 3-shot prompting.*
>
> - **Response:** In Appendix F in the supplement (Figure 7), we experiment with other models, including Gemma 1 2B and Gemma 7B.
>
> > For the filter-only setup, is the model trained on less samples (than w/ boosting) since we filter the bad ones out?
>
> No, in all settings considered, there are more correct examples than (training steps * batch size). The difference between the setups is not in the quantity of examples, but the source (either from the weak learner or from the model itself).

---

### Official Review · Reviewer_QYsv · 2025-06-28

**Clarity:** 3
**Significance:** 3
**Originality:** 3
**Rating:** 5
**Confidence:** 3

**Summary:**

This paper addresses "model collapse" in LLMs trained on synthetic data by proposing a theoretical framework inspired by "boosting". It argues a "strong learner" (a powerful LLM) can continuously improve using only "weak data"—a minimal, external signal. The proposed algorithm iteratively identifies prompts where the model fails ("hard examples") and focuses a "weak labeler" on them. This feedback is mixed with high-quality synthetic data for retraining. The work proves this method prevents collapse and ensures convergence to an optimal model, outperforming baselines in experiments.

**Questions:**

N/A

**Ethical Concerns:**

["NO or VERY MINOR ethics concerns only"]

**Final Justification:**

My concerns are resolved.

**Limitations:**

Yes

**Quality:**

3

**Strengths And Weaknesses:**

Strengths:
1. The paper addresses the critical and urgent problem of "model collapse" when training on synthetic data for the sustainable development of Large Language Models (LLMs).
2. The interesting theoretical framework offers a deep and insightful explanation for why the method is effective.
3. The experiments are well-structured to validate the theoretical claims.

Weakness:
+ The framework assumes a binary (good/bad), unambiguous, and efficiently evaluable quality function q. This is true for the tested tasks of math and coding, but for more subjective tasks like creative writing or summarization, quality is nuanced and hard to define, limiting the direct applicability of the theory.

---

> ### Author Rebuttal · Authors · 2025-07-31
>
> We thank the reviewer for their positive assessment of our paper. We appreciate their recognition that (paraphrasing from their response) our paper addresses a critical and urgent problem, offers a deep and insightful explanation of our method, and contains experiments which validate our theoretical claims.
>
> We respond to their main concern below.
>
> **Weakness:** *The framework assumes a binary (good/bad), unambiguous, and efficiently evaluable quality function q. This is true for the tested tasks of math and coding, but for more subjective tasks like creative writing or summarization, quality is nuanced and hard to define, limiting the direct applicability of the theory.*
>
> - **Response:** In fact, we relax this assumption in Appendix A.2 in the supplement, where we show that our results straightforwardly extend to a setting where filtering according to the quality function can only be performed imperfectly.

---

> > ### Comment · Reviewer_QYsv · 2025-08-04
> >
> > Thank you for your response! I will not update my scores.

---

### Official Review · Reviewer_Ckek · 2025-07-02

**Clarity:** 2
**Significance:** 2
**Originality:** 2
**Rating:** 2
**Confidence:** 4

**Summary:**

The paper formalizes “training on synthetic data with minimal curation” as a boosting-style process. Experiments fine-tune Gemma-2 2B on GSM8K and MBPP. “Boosting” improves training accuracy and slightly increases test accuracy compared with (a) “do nothing” and (b) “filter-only” baselines, but the gains over the best supervised baseline are small

**Questions:**

1. In practice the learner minimizes cross-entropy for limited epochs. How sensitive is convergence if the model only approximates the empirical distribution?
2. What is the wall-clock time and GPU budget per round for k = 8 on GSM8K? How would this scale to a 70B model on 1M prompts?
3. Does the algorithm still converge if the labeler’s errors are systematic (biased) rather than random?

**Ethical Concerns:**

["NO or VERY MINOR ethics concerns only"]

**Limitations:**

See above.

**Quality:**

2

**Strengths And Weaknesses:**

Strengths:
* Unifying formalism that connects recent self-improvement heuristics (STaR / ReST) to classical boosting.
* Empirical section reproduces the model-collapse phenomenon and shows the proposed variant can mitigate it.

Weakness:
* The proof requires every generation to exactly fit the weighted dataset (Definition 1). This is unattainable for billion-parameter LLMs trained for a few hundred steps, undermining the practical relevance of Theorem 6.
* Experiments simulate the labeler with an already pretrained Gemma 2B, i.e. they assume access to an LLM stronger than the initial model. No evidence is given that a genuinely weak.
* No error bars, confidence intervals, or significance tests are reported.

---

> ### Author Rebuttal · Authors · 2025-07-31
>
> We thank the reviewer for their engagement with our paper. We appreciate their recognition that (paraphrasing from their review) we provide a unifying formalism that connects recent self-improvement heuristics (STaR / ReST) to classical boosting, and that our experiments reproduce the model-collapse phenomenon while showing how our proposed variant can mitigate it.
>
> We respond to their concerns and questions below, **nearly all of which seem to be based on misunderstandings**:
>
> **Weakness:** *Boosting improves training accuracy and slightly increases test accuracy compared with (a) do nothing and (b) filter-only baselines, but the gains over the best supervised baseline are small.*
>
> - **Response:** In fact, the gains are large. Table 1 (“GSM8k test” column) shows that boosting leads to substantial improvements in test accuracy over (1) do nothing baseline (29.3% absolute, 191.5% relative); (2) filter-only baseline (7.5% absolute, 20.2% relative); (3) supervised baseline, i.e., Gold SFT (7.7% absolute, 20.9% relative).
>
> **Weakness:** *Experiments simulate the labeler with an already pretrained Gemma 2B, i.e. they assume access to an LLM stronger than the initial model. No evidence is given that a genuinely weak...*
>
> - **Response:** In fact, none of our experiments use a labeler that is stronger than the initial model. In Section 7, both the initial model and labeler are Gemma 2 2B PT, which gets a test accuracy of 21.9%. In Appendix F in the supplement (Figure 7), the initial model is also Gemma 2 2B PT, but the labeler is Gemma 1 2B PT, which gets a test accuracy of 11%. We see improvements across rounds compared to the initial model (red dotted line). This demonstrates the success of our approach with genuinely weak labeler.
>
> **Weakness:** *No error bars, confidence intervals, or significance tests are reported.*
>
> - **Response:** We do report them. In Appendix F in the supplement (Table 4 and Figure 5) we report mean and standard deviation for all results in the main text (Table 1 and Figure 1). We ran 3 trials per experiment, which is more trials than the STaR/ReST_EM papers.
>
> **Weakness/Question:** *The proof requires every generation to exactly fit the weighted dataset (Definition 1) ... How sensitive is convergence if the model only approximates the empirical distribution?*
>
> - **Response:** While our theory assumes an idealized strong learner, in our experiments we used a real learner instead of a strong learner, and our algorithm outperformed several baselines. So we think we have demonstrated that our theory is practically relevant, as it led to the development of a high-performing algorithm. Furthermore, our assumption is motivated by the widely observed phenomenon that LLMs memorize their training data. We agree that further weakening of the assumption would be an interesting extension of our theory, and view spurring research in that direction as a primary benefit of publishing our paper.
>
> **Question:** *Does the algorithm still converge if the labeler’s errors are systematic (biased) rather than random?*
>
> - **Response:** We do not assume that the labeler’s errors are random. In Section 4.2 (line 181) we say that they can be arbitrary.
>
>
> **Question:** What is the wall-clock time and GPU budget per round for k = 8 on GSM8K?
>
> - **Response:** For GSM8K, and k=8, we sample 112k responses each round, which dominates the accelerator time compared to training, and requires approximately ~16 H100-equivalent hours. Scaling up to a 35x larger model and 142x larger dataset would suggest a 5000x increase in required compute, applying the flops = 2 * params * tokens estimate (assuming inference is batched enough to be compute bottlenecked).

---

> > ### Author Response · Authors · 2025-08-06
> > **Response?**
> >
> > We thank the reviewer again for their engagement with our paper. We hope that our rebuttal addresses their concerns. If there are any follow-up questions, we would be happy to answer them.

---

### Official Review · Reviewer_drWH · 2025-07-05

**Clarity:** 3
**Significance:** 3
**Originality:** 3
**Rating:** 4
**Confidence:** 3

**Summary:**

This paper presents an effort to place the popular practice of LLM self-improvement—through rounds of synthetic data generation/pseudo-labelling, filtering, and fine-tuning—within a unifying theoretical framework. By inverting the traditional boosting framework from learning theory—shifting from weak learners with strong data to a setting with a strong learner and weak data, reflective of the current state of LLMs—they introduce a boosting-style algorithm that, upon convergence, offers probabilistic guarantees of producing correct responses to nearly all prompts. Experimental results display the effects of the combined algorithmic components on train and test error on mathematical and coding benchmarks, while further experiment in the appendix aims to empirically assess some relaxations of assumptions to the framework.

**Questions:**

- The impact of boosting towards the achievable test error remains unclear. Even though this is not set forth as a core concern of the paper it would be helpful for the reader that the authors seek a better explanation of the observed results. Moreover, can removing the focus on hard examples allow faster improvement over the first few rounds (as hinted in Fig.1 left), hence potentially be preferable under smaller computation budgets?

- I think a valuable addition would be to explicitly compare the existing frameworks mentioned in the paper (STAR, REST, REST-EM) with the proposed framework, highlighting missing components or key variations. Moreover, including these frameworks as part of the experiments would provided useful baselines and expose which stages of the algorithm result in making a difference in practice.

- A useful addition would be some further studies on the impact of the involved hyperparameters (e.g., the number of samples $k$, and fractions $\alpha$ and $\beta$).

**Ethical Concerns:**

["NO or VERY MINOR ethics concerns only"]

**Final Justification:**

Thank you for following up. My assessment remains the same based on the provided clarifications.
My main remaining point is the paper to be revised towards a better integration of the main part with the considerations/findings on weakening the assumptions for the learner and filtering, already attempted in the appendix.

**Limitations:**

Yes

**Quality:**

3

**Strengths And Weaknesses:**

**Strengths**

The paper comes up with an insightful theoretical framework for self-improvement and some theoretically-backed design principles for further developments in the area. The proposed boosting style algorithm is general enough to encompass commonly used frameworks and provides one of the first guarantees of convergence for self-improvement. Moreover, it allows diagnosing weaknesses of existing frameworks.

**Weaknesses**

The work rests on some strong assumptions for the involved LLMs. Most notably the derivation assumes that the LLM can be considered as a strong learner powerful enough to interpolate a given conditional probability (which might be unrealistic in practice). Moreover, a perfect filter is also assumed. Thirdly, the verifier is considered sufficiently powerful to extract a (potentially tiny) fraction of high quality labels, which can be realized programmatically for some learning task categories, however again this might be facing limits in the wild. Empirical findings provide a mixed signal for the advantage that focusing on hard examples gives.

---

> ### Author Rebuttal · Authors · 2025-07-31
>
> We thank the reviewer for their positive assessment of our paper. We appreciate their recognition that (paraphrasing from their review) our algorithm is general enough to encompass existing frameworks, enabling diagnosis of their weaknesses, and provides one of the first guarantees of convergence for self-improvement.
>
> We address their concerns and questions below:
>
> **Weakness:** *The work rests on some strong assumptions for the involved LLMs. Most notably the derivation assumes that the LLM can be considered as a strong learner powerful enough to interpolate a given conditional probability (which might be unrealistic in practice).*
>
> - **Response:** While our theory assumes an idealized strong learner, in our experiments we used a real learner instead of a strong learner, and our algorithm outperformed several baselines. So we think we have demonstrated that our theory is practically relevant, as it led to the development of a high-performing algorithm. Furthermore, our assumption is motivated by the widely observed phenomenon that LLMs memorize their training data. We agree that further weakening of the assumption would be an interesting extension of our theory, and view spurring research in that direction as a primary benefit of publishing our paper.
>
> **Weakness:** *A perfect filter is also assumed.*
>
> - **Response:** In fact, we relax this assumption in Appendix A.2 in the supplement, where we show that our results straightforwardly extend to a setting where the filter is imperfect.
>
> **Question:** *I think a valuable addition would be to explicitly compare the existing frameworks mentioned in the paper (STAR, REST, REST-EM) with the proposed framework.*
>
> - **Response:** In fact, we explicitly compare to STaR/ReST. See line 275.
>
> **Question:** *Can removing the focus on hard examples allow faster improvement over the first few rounds (as hinted in Fig.1 left), hence potentially be preferable under smaller computation budgets?*
>
> - **Response:** We agree with the reviewer that the benefit of focusing on the hard examples appears to mostly be realized in the later rounds of our algorithm, and thus might only be worthwhile when computational resources are adequate.
>
> **Question.**: A useful addition would be some further studies on the impact of the involved hyperparameters (e.g., the number of samples $k$, and fractions $\alpha$ and $\beta$).
> - **Response:** As $\beta$ is a quantity determined by the quality of the weak labeler, we implicitly vary this parameter in our *off-policy* experiments in Figure 7 of the Appendix, where we experiment with different labelers and find faster improvement with larger $\beta$ as the theory predicts.

---

### Note · Authors · 2025-08-11

We thank the reviewers for their time. We believe that many of the questions and concerns they raised are addressed in the main body of the paper or in the appendix, and we refer the reviewers to our detailed rebuttals for pointers to the relevant sections.

---

### Decision · Program_Chairs · 2025-09-17

**Decision:**

Accept (poster)

**Comment:**

Paper studies the use of synthetic data to iteratively improve an LLM. Specifically they study how much curation needs to be applied on the synthetic data for the model to not collapse and continually improve. They study an algorithm which subsumes known successful methods and provide a novel analysis inspired by classical boosting literature. Under certain assumptions they show that focusing the synthetic labeling on challenging samples leads to continual improvement. These allow the authors to propose improvements to existing methods.

Most reviewers considered the contributions important, novel and well supported by promising empirical results. Reviewer drWH and Ckek pointed out that the assumptions are too strong, but the authors responded that their empirical performance is evidence that their theory is useful. Reviwer QYsv brought up that proposed method may only work in domains like math and coding where the correctness of a label is exact. Authors point to Appendix A.2 which relaxes this assumption in theory, but doesn't any provide empirical validation. Reviewer BnJF raises that performance drops for T>5 number of model iterations, but authors do not provide any explanation. Authors are encouraged to provide more experiments with diverse set of datasets and longer iterations in the next revision.